# An Integrative Neuromuscular Training Program in Physical Education Classes Improves Strength and Speed Performance

**DOI:** 10.3390/healthcare13121372

**Published:** 2025-06-08

**Authors:** Diego A. Alonso-Aubin, Ignacio Moya del Saz, Ismael Martínez-Guardado, Iván Chulvi-Medrano

**Affiliations:** 1Strength Training and Neuromuscular Performance Research Group (STreNgthP), Faculty of Health Sciences—HM Hospitals, University Camilo José Cela, C/Castillo de Alarcón, 49, Villanueva de la Cañada, 28692 Madrid, Spain; 2HM Hospitals Health Research Institute, 28015 Madrid, Spain; 3Faculty of Health Sciences—HM Hospitals, University Camilo José Cela, C/Castillo de Alarcón, 49, Villanueva de la Cañada, 28692 Madrid, Spain; nachomds2@hotmail.com; 4LFE Research Group, Department of Health and Human Performance, Faculty of Physical Activity and Sport Science (INEF), Universidad Politécnica de Madrid, C/Martín Fierro, 7, 28040 Madrid, Spain; ismael.mguardado@upm.es; 5Sport Performance and Physical Fitness Research Group (UIRFIDE), Facultad de Ciencias de la Actividad Física y el Deporte, Universidad de Valencia, 46010 Valencia, Spain; ivan.chulvi@uv.es

**Keywords:** resistance training, power training, youth training, school-based training

## Abstract

**Background/Objectives:** This longitudinal randomized controlled study aimed to evaluate the effects of an integrative neuromuscular training (INT) intervention on strength and speed measures in Spanish students across different age groups. **Methods**: A total of 121 students, aged 11–12 (G1) and 15–16 (G2), were randomly assigned to four groups: two experimental groups, G1exp (n = 30) and G2exp (n = 31), and two control groups, G1con (n = 30) and G2 (n = 30). Experimental groups participated in two 20 min INT sessions per week for four weeks, focusing on physical literacy (agility, coordination, balance, and speed) and a resistance training program, integrated into the first part of physical education classes (PEC). **Results**: Experimental groups showed significant improvements (*p* < 0.001) in lower-body power (G1exp: t = −7.04; d = −1.30; G2exp: t = −5.19; d = −0.91), upper-body power (G1exp: t = −5.94; d = −1.10; G2exp: t = −3.52; d = −0.62), abdominal endurance strength (G1exp: t = −9.72; d = −1.80; G2exp: t = −4.75; d = −0.84) and sprinting (G1exp: t = 5.22; d = 0.96; G2exp: t = 5.90; d = 1.04). A comparison by age groups revealed significantly greater improvements in upper-body power in G1 vs. G2 (t:−2.83; *p* = 0.02). **Conclusions**: A four-week INT program implemented at the start of PE classes can improve strength and sprint performance in youth across all ages. We recommend incorporating INT into the first part of physical education sessions to contribute to meeting the physical activity, well-being and conditioning needs of young people.

## 1. Introduction

The relationship between physical activity and physical fitness in children and adolescents is a crucial research area, with significant implications for health and well-being [1]. Physical activity (PA) is consistently associated with various aspects of physical fitness, including muscular strength, endurance, flexibility, and cardiovascular health, and encompasses daily routine activities [2]. Physical fitness serves as key indicator of the body’s capacity for PA and overall health [3,4].

However, PA levels are steadily declining in most countries worldwide [5]. Physical inactivity, defined as insufficient engagement in PA that fails to meet current recommendations, has been identified as the fourth leading risk factor for premature mortality in adulthood [6,7].

Current guidelines recommend at least 60 min of moderate to vigorous physical activity (MVPA) daily during childhood and adolescence, including at least three days of resistance training per week [8]. However, PA declines during childhood and adolescence, with a notable drop in MVPA around age 11, highlighting this stage as critical for intervention [9,10]. Additionally, young people have shown measurable declines in muscular strength, power, and motor skill performance, alongside rising rates of overweight and obesity [11]. This can lead to the pediatric triad: exercise deficit disorder, pediatric dynapenia, and physical illiteracy [12]. The pediatric triad is associated with a decrease in activity levels, falling below the recommended 60 min of MVPA, a reduction in the ability to generate high-strength and -power muscle actions, and the inability to perform coordinated actions based on fundamental motor skills. Understanding these trends is essential for developing effective strategies to promote a healthier lifestyle, meeting PA guidelines and MVPA durations, early in life [13,14].

Integrative neuromuscular training (INT) has been proposed as an effective approach to improving the pediatric triad. INT is a conditioning program that combines general and specific strength and conditioning exercises, including resistance training, agility, balance, coordination, and speed drills, to enhance muscle strength and physical literacy [15]. INT-routines can be performed in 10- to 20 min sessions and integrated into sports programs or physical education classes (PEC) [16,17].

PECs play a crucial role in promoting PA during childhood and adolescence due to their mandatory inclusion in educational programs [18,19]. PECs also help develop motor competence and physical fitness levels, fostering confidence to engage in both structured and spontaneous MVPA [20,21]. However, PEC curricula often lack strength and conditioning content, as they have traditionally focused on team sports and games [22]. These activities may not adequately prepare young people for a lifetime of physical activity, and may fail to engage the least active and least skilled young people [23].

On the one hand, school-based neuromuscular training programs may offer greater benefits to PEC alone in enhancing postural control, fundamental movement skills, and muscular strength [24]. However, few studies have examined their effects. The existing research has shown positive outcomes, including improvements in fundamental movement skill in children aged 6–7 years [25], postural control and flexibility [26,27], endurance [17], and strength [28]. Nevertheless, more evidence is needed due the variability in participant characteristics across studies, as there is limited research on school-based intervention programs targeting different age groups. Previous studies have considered longer intervention times using training materials. However, the reality sometimes forces exercise professionals and educators to work with limited resources, both in terms of materials and time.

In this sense, this study aimed to address gaps in the literature regarding the implementation of INT in PEC for high-school students (preadolescents and adolescents). The objective was to evaluate the effects of four-week INT intervention on strength and speed measures in Spanish high-school students. It was hypothesized that the groups undergoing the INT intervention would show greater improvements in physical performance, specifically in lower-body power, upper-body power, abdominal endurance strength, and sprinting, compared to the control groups.

## 2. Materials and Methods

This longitudinal randomized controlled study was conducted in a Spanish high school during PEC program over a period of six weeks, divided into three phases. The first and last weeks were dedicated to data collection and testing, while the four weeks between the second and fifth weeks were used for an intervention involving INT, during which high-school students trained twice a week. The inclusion criteria were as follows: (1) no previous participation in an INT program, performing all the intervention classes, and an ability to perform all the intervention classes; and (2) any musculoskeletal injury within the past year, or any medical condition (physical or mental) that could limit exercise performance. The exclusion criteria included: (1) failure to attend at least 85% of the training sessions, and (2) consumption of any stimulant beverage prior to the performance tests (e.g., energy drinks, caffeine, etc.).

All participants, along with their parents or guardians, were informed about the study’s purpose, experimental procedures, and potential risks. They were given the opportunity to ask any questions regarding the procedures. After receiving this information, all participants signed a child assent form, and all parents or legal guardians signed a parental permission form. All the procedures of this study were performed in accordance with the Declaration of Helsinki and approved by the Ethics Committee of the Camilo José Cela University (06_25_ENTCF).


**Participants**


One hundred and twenty-one high-school students were selected for participation and randomly assigned to four groups. To randomize group assignment, we used Microsoft Excel^®^ using the randomize function, and then sorted the participants into the four groups for intervention or control. For the intervention, two different secondary school grades were selected: 1st grade (G1) (ages 11–12) and 4th grade (G2) (ages 15–16). Within each grade, two groups were formed: one experimental group, divided into G1exp (n = 30) and G2exp (n = 31), and one control group, divided into G1con (n = 30) and G2 (n = 30). The experimental groups participated in both PEC and the INT program, while the control groups only attended the PEC.


**Measurements and procedures**


All the participants underwent a familiarization week in which they practiced all the tests. The first day of both the first and sixth weeks was dedicated to assessments. The main researcher, an experienced strength and conditioning coach as well as a physical education teacher, oversaw the testing and intervention process, which was conducted in the same facilities and followed a consistent time schedule. Additionally, another researcher collected the data while blinded to the group assignments. Students were advised to avoid vigorous physical activity for 24 h prior to the testing session.

General testing procedures were followed in both assessment weeks. First, students completed a five-minute dynamic warm-up, which included joint mobility exercises, a three-minute low-intensity run, ten squats, five push-ups, and one 30 m forward run at medium intensity. After the warm-up, students performed the following assessments in the following order, with a two-minute rest between each test: standing long jump, medicine ball throw (3 kg), 30 s crunch test, and 4 × 10 m sprint. These tests are widely used in the literature due to their strong validity and reliability [29].


**
*Lower-body power: standing long jump.*
**


The student stood behind the starting line with their feet shoulder-width apart, ensuring that their heels remained on the ground before takeoff. The jumping technique was not corrected during the execution. Performance was measured by recording the distance from the starting line to the point where the heels landed. Each participant performed two jumps, and the best one was recorded.


**
*Upper-body power: medicine ball throw (3 kg).*
**


The students were placed in a sitting position with their knees fully extended, feet together, and heels positioned on the starting line, and were not permitted to lift them from the ground at any time. Holding a medicine ball (3 kg) close to the chest, the student executed a forward throw by rapidly extending the elbows. The throw distance was measured from the starting line to the point where the ball landed. Each participant performed two throws, and the best attempt was recorded.


**
*Abdominal endurance strength: 30 s crunches.*
**


The student lay in a supine position with their knees bent at a 90° angle and ankle firmly on the floor. The researcher assisted by holding the student’s ankles in place. The arms were positioned behind the neck, with elbows flexed forward. A crunch was considered valid when the student touched their knees with their elbows. Each participant performed one set and as many crunches as possible in 30 s, with only valid repetitions being recorded.


**
*Sprint: 4 × 10 m round trip.*
**


The student stood in front of the starting line and completed a series of round-trip sprints over a 10 m distance, totaling 80 m at the highest possible speed. The test began as soon as the student initiated movement, which triggered the stopwatch. At each turn, the student had to cross the 10 m line with both feet. The test concluded when the student stepped beyond the finish line with both feet. Each participant performed one sprint, and the completion time was recorded.


**Integrative Neuromuscular Training (INT)**


After the initial week of assessments, students began the intervention period, which consisted of two sessions per week over a duration of four weeks. Students in the G1con and G2con groups, following a general warm-up, completed a traditional specific warm-up that included mobility exercises and basic stretching. In contrast, students in the G1exp and G2exp groups participated in 20 min of INT, based on the program outlined in Table 1.


**Physical Education Classes (PEC)**


After completing either the INT session (G1exp and G2exp) or the warm-up (G1con and G2con), all four groups participated in 35 min of PEC, which consisted of game-based activities. These games incorporated changes in direction, running at different speeds, and manipulative skills.


**Statistical Analysis**


All dependent variables are presented as means ± standard deviation (SD). Before analysis, data normality and homogeneity of variance were assessed with the Shapiro–Wilk and Levene’s tests, respectively. A paired *t*-test was used to evaluate within-group changes over time. Additionally, effect sizes (ES) were calculated using the Cohen’s d, classified as small (d = 0.00 to ≤0.49), medium (d = 0.50 to ≤0.79), and large (d = ≥0.80) [30]. One-way ANOVA was conducted with Fisher correction and for groups comparisons followed by Tukey’s post hoc analysis test to identify significant differences.

The statistical power of the study was assessed using G*Power (version 3.1.9.7) for an analysis of variance (ANOVA) with a moderate effect size (Cohen’s d = 0.50), a significance level (α) of 0.05, and a desired power of 0.80. With a total of 121 participants distributed across four groups, the calculated statistical power was 99.98%, ensuring a high probability of identifying a moderate effect if present.

For all statistical tests, a significance level of *p* ≤ 0.05 was applied. All data analyses were performed using SPSS 22.0 (SPSS, Inc, Chicago, IL, USA).

## 3. Results

All subjects completed the testing procedures and the INT program with no unexpected events or injuries during the study period. All variables were normally distributed and showed homogeneity of variance.

Significant intragroup differences were found in the following groups and dependent variables: G1exp (lower-body power: *p* < 0.001, t = −7.94; d = −1.30; higher body power: *p* < 0.001, t = −5.94; d = −1.10; abdominal endurance strength: *p* < 0.001, t = −9.72; d = −1.80 and sprint: *p* < 0.001, t = 5.22; d = 0.96), G2exp (lower-body power: *p* < 0.001, t = −5.19; d = −0.91; upper-body power: *p* < 0.001, t = −3.52; d = −0.62; abdominal endurance strength: *p* < 0.001, t = −4.75; d = −0.84; and sprinting: *p* < 0.001, t = 5.90; d = 1.04), and G1con (abdominal endurance strength: *p* < 0.001, t = −5.21; d = −0.96). All data are shown in Table 2.

Significant group differences were found in lower-body power–previous INT (G1exp vs. G2con: t = −3.11, *p* = 0.01), lower-body power–posterior INT (G2exp vs. G1con: t = 2.64; *p* = 0.04), upper-body power–previous INT (G1exp vs. G2exp: t = −3.72, *p* = 0.005; G1exp vs. G2con: t = −5.91, *p* = 0.001 and G1con vs. G2con: t = −4.75, *p* < 0.001), upper-body power–posterior INT (G1exp vs. G2exp: t = −2.83, *p* = 0.02; G1exp vs. G2con: t = −4.98, *p* < 0.001; G2exp vs. G1con: t = 2.71, *p* = 0.03 and G1con vs. G2con: t = −4.87, *p* < 0.001), abdominal endurance strength–previous INT (G1con vs. G2con: t = −2.85, *p* = 0.02), abdominal endurance strength–posterior INT (G2exp vs. G1con: t = −4.15, *p* < 0.001 and G1con vs. G2con: t = −2.59, *p* = 0.05), sprint–previous INT (G1exp vs. G1con: t = −2.81, *p* = 0.02 and G1con vs. G2con: t = −3.73, *p* = 0.002), and sprint–posterior INT (G1exp vs. G1con: t = −4.73, *p* < 0.001; G2exp vs. G1con: t = −5.07, *p* < 0.001 and G1con vs. G2con: t = 4.30, *p* < 0.001). All data shown in Table 3 and Figure 1 and Figure 2 show pre–post differences in G1exp (puberal) and G2exp (adolescents).

In Figure 1, we can observe the differences obtained in the G1exp (ages 11–21) between the pre- and post-intervention measurements. The differences were statistically significant in the standing long jump, medicine ball throw, and sprint. *p* denotes significance index (*: *p* ≤ 0.05; **: *p* ≤ 0.01).

In Figure 2, we can observe the differences obtained in the G2exp (ages 15–16) between the pre- and post-intervention measurements. The differences were statistically significant in the standing long jump, medicine ball throw, abdominal endurance strength, and sprint. *p* denotes significance index (*: *p* ≤ 0.05; **: *p* ≤ 0.01).

## 4. Discussion

The aim of the present study was to determine the effects of an INT (integrated neuromuscular training) intervention on strength and speed outcomes in Spanish secondary school students across different academic grades. The results demonstrate that an INT program implemented twice per week over a four-week period can significantly improve strength and sprint performance in adolescents, regardless of whether they are in preadolescence or adolescence.

It is important to highlight that the groups that performed the INT intervention showed improvements across all the variables assessed, whereas the control group (G1con) only demonstrated a significant improvement in abdominal endurance strength. When comparing age, significant differences were found between G1exp and G1con in both abdominal endurance strength and sprint performance. In contrast, no significant differences were observed between G2exp and G2con.

INT programs are thus proposed as a valuable resource and alternative for implementing physical activities and exercises, particularly within PEC [17,31]. These programs may contribute to meeting the physical activity and conditioning needs of young people [18,32].

One of the major public health concerns today is the low levels of physical activity and fitness among young people. In this regard, it is recommended that children and adolescents engage in at least 60 min of moderate-to-vigorous physical activity daily [33]. However, despite this recognized need, there remains a lack of institutional clarity regarding the specific content and structure of physical activity during this time [15]. Current guidelines generally suggest performing a minimum of 60 min of aerobic activity per day, along with muscle-strengthening exercises at least three times per week [34].

The implementation of INT programs in the school setting appears to be a promising strategy for promoting muscular fitness and its associated health benefits. Therefore, further research in this area is warranted to better understand its potential and optimize its application in youth populations [35,36]. Recent review suggested the need to implement at least three sessions per week to elicit moderate improvements in muscular endurance and greater gains in muscular strength and power [36].

Our results are consistent with previous studies that have implemented INT programs in other contexts [37]. For example, a study conducted with youth rugby players reported similar outcomes following an eight-week INT intervention, with significant improvements in lower-body power, upper-body power, abdominal endurance strength, and sprint performance [16]. In tennis players, INT programs performed twice per week over eight consecutive weeks led to enhancements in sprint performance and changes in direction ability [38]. Additionally, a study involving female badminton players found that an eight-week INT program improved motor skills, reduced limb asymmetry, and contributed to injury prevention and performance enhancement [39].

In the studies mentioned above, participants generally had a higher baseline level of physical activity, as they were already engaged in structured exercise within a sports environment. In contrast, our participants had a more heterogeneous and overall lower initial fitness level. Despite this, our results showed greater improvements, suggesting that individuals with lower baseline fitness may benefit more markedly from INT interventions. Furthermore, while previous studies implemented interventions over eight weeks, the present study achieved significant improvements with a shorter, four-week program. These findings indicate that substantial gains in physical fitness can be achieved in both athletic and general youth populations, supporting the applicability of integrated neuromuscular training across diverse contexts, and thereby promoting overall health and well-being. Muscular fitness is particularly important due to its broad range of health-related benefits, including favorable associations with total and central adiposity, cardiovascular and metabolic risk factors, and bone health, as well as psychological outcomes such as self-esteem and perceived sport competence [40,41]. Moreover, muscular fitness may be associated with the maintenance of key health parameters later in life, contributing to healthier aging trajectories.

The G1exp group shows larger effect sizes than the G2exp group. Puberty and the prepubertal stage exhibit greater neuroendocrine and likely, neuromuscular plasticity, requiring less stimulus to induce changes compared to late adolescence. Interventions during these stages may be more effective due to this increased sensitivity [15,42].

On the one hand, a few studies have implemented INT programs within physical education settings. One reason for the limited research in this context is the concern regarding the potential risk associated with implementing training programs of this nature [43]. However, evidence from a school-based neuromuscular training program has shown a reduction in injury incidence [44]. In line with these findings, no injuries were reported in our study. Therefore, it is recommended that further research be conducted in schools and educational institutions, as INT programs can be safely implemented during childhood [42].

Regarding physical qualities, our results indicate that an INT program implemented within PEC can effectively improve strength and sprint performance. Similar findings were reported in a school-based systematic. In a 10-week INT program involving children aged 6–7 years, significant improvements were observed in fundamental movement skills [25]. These gains may be attributed to neural and muscular adaptations driven by the high plasticity of the neuromuscular system during early development [45,46].

In this context, resistance training has been shown to enhance muscular adaptations that improve both upper- and lower-body strength and power in youth populations [47,48,49,50]. Moreover, increases in strength and power, along with the continued application of resistance training, have been associated with improved motor skill competence in adolescents [51]. This may enable greater motor competence in young people, allowing them to achieve appropriate levels of both quantity and quality of physical activity, thereby improving their physical, emotional, social, and psychological health and well-being [52].

Our results demonstrate that significant improvements can be achieved in both upper- and lower-body strength and power performance through INT interventions. Similar outcomes were reported by Kennedy et al. (2018), who found immediate and sustained improvements in upper-body muscular fitness and skill competency following a six-month resistance training program in adolescents [23]. Likewise, Katsanis et al. (2021) observed notable gains in strength and power after an eight-month resistance training program, performed twice weekly, utilizing suspension training methods [53]. Furthermore, resistance training programs incorporating machines, free weights, and plyometric exercises have been shown to positively impact muscular fitness in adolescent populations [35].

Upon observing the results in the experimental groups (G1exp and G2exp), significant improvements are seen in all areas except abdominal endurance strength (G1exp), with large effect sizes. This implies that the INT programs provide substantial quantitative and qualitative benefits in physical fitness and health for young individuals, regardless of their age group.

In addition, incorporating strength exercises into PEC in secondary schools has been shown to improve body composition, particularly through increases in fat-free mass [54]. Therefore, it is recommended to integrate a periodized resistance training program into the school curriculum as a strategy to promote the development of athleticism and overall physical health in all young people [55].

Abdominal endurance strength can be effectively developed in young people, contributing to improved posture, balance, and stability, all of which positively influence motor skills development in children [56]. In our study, both the experimental and control groups achieved significant improvements in abdominal endurance strength, likely due to the inclusion of core exercises within the physical education curriculum. However, the experimental groups demonstrated significantly greater improvements compared to the control groups. These findings align with previous research reporting increases in abdominal endurance strength following a 16-week intervention combining INT and yoga [57].

The differences in the development of strength and abdominal endurance between preadolescents and adolescents may be influenced by factors such as physical development, physical activity, and training methods. Adolescents tend to show less change in abdominal endurance than preadolescents, possibly due to differences in maturation, motivation, and training adaptation [58].

On the one hand, sprint performance showed substantial improvements in both experimental groups following the INT intervention. This may be attributed to the well-established relationship between gains in strength and power and enhancements in sprint performance [59]. Supporting this, a study involving high-school students reported significant improvements in sprint ability after an eight-week sled-towing program conducted twice weekly, compared to standard physical education classes [60]. Our findings suggest that implementing an INT program within physical education may yield comparable benefits while requiring less time commitment.

Although cardiometabolic changes were not assessed in our study, previous research has reported improvements in peak oxygen uptake (VO_2_), suggesting that INT programs can provide a moderate-to-vigorous cardiometabolic stimulus. In fact, such programs may be metabolically demanding, potentially even more than treadmill walking [61].

Finally, in the pairwise comparison between G1exp and G1con (11–12 years old), significant differences were found in both abdominal endurance strength and sprint performance. However, no significant differences were observed in the comparison between G2exp and G2con (15–16 years old). This may be explained by developmental differences: during adolescence, greater training stimuli (e.g., increased duration, volume, or intensity) may be required to elicit measurable improvements in physical performance variables. In contrast, pre-adolescent individuals may exhibit greater biological sensitivity and responsiveness to the same stimulus due to their stage of neuromuscular development.

Some limitations of this study include not being able to precisely measure the activity level of the groups, although efforts were made to minimize the performance of activities not included at the start of the intervention. Additionally, it would have been interesting to follow up on the results to determine the long-term effect of the findings and/or implement a counterbalance. On one other hand, it would have been interesting to collect the amount of exercise performed by all participants before starting the study to standardize the activity level and make evaluations based on the prior activity level.

Future research should conduct longer intervention programs across different developmental stages to understand the impact of implementing intervention programs. It would be especially interesting to categorize groups based on the peak growth spurt in height to determine the time of the greatest beneficial effects in these types of programs.

However, this study provides very interesting results, as significant improvements have been observed with a short intervention period and the use of a methodology that does not require sophisticated materials. Additionally, the study was conducted during the school term, which will allow teachers to manage these interventions properly without compromising their curricular content.

## 5. Conclusions

These findings indicate that a four-week INT program can effectively improve lower- and upper-body strength, abdominal endurance strength, and sprint performance when implemented during the initial phase of physical education classes, regardless of students’ grade or age. Based on these results, we recommend incorporating INT into the first part of physical education sessions for both preadolescent and adolescent students to contribute to meeting the physical activity, well-being, and conditioning needs of young people.

Despite knowing the benefits of INT in these population groups, some practical challenges should be mentioned that professionals must consider: (1) structuring their sessions to include INT programs with a time interval of 10–15 min, (2) understanding the material limitations that may affect exercise progression, (3) in educational institutions, it would be necessary to provide training for safe and effective implementation, and (4) considering the maturation and developmental status of the participants.

## Figures and Tables

**Figure 1 healthcare-13-01372-f001:**
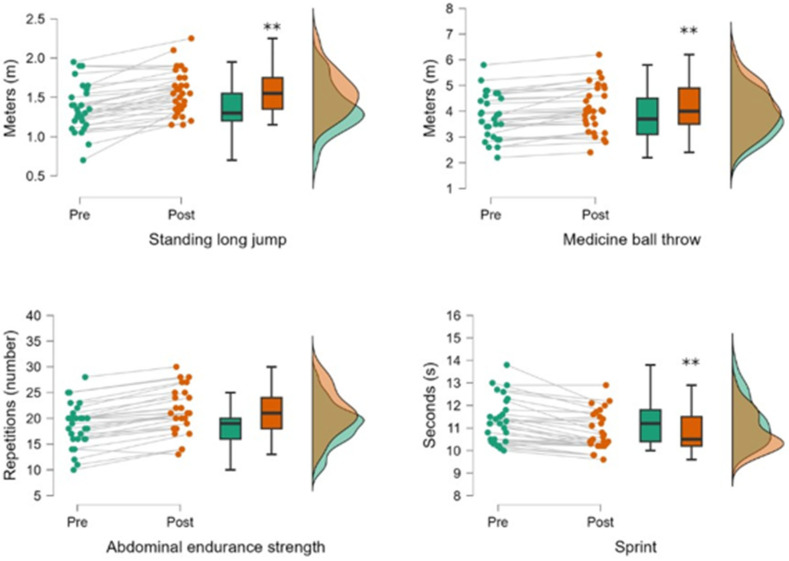
Differences in physical outcomes in G1exp (**: *p* ≤ 0.01).

**Figure 2 healthcare-13-01372-f002:**
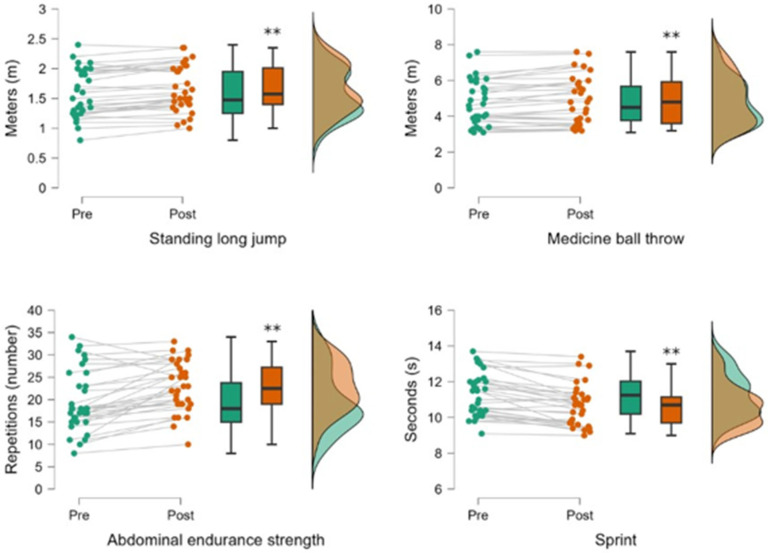
Differences in physical outcomes in G2exp (**: *p* ≤ 0.01).

**Table 1 healthcare-13-01372-t001:** Integrative neuromuscular training.

	Session 1		Session 2	
Weeks	Exercises	Volume (sets × reps and recovery) and intensity	Exercises	Volume (sets × reps and recovery) and intensity
Week 1	Hip mobility	2 × 10; R:15 s; BW	Hamstring mobility	2 × 5 + 5; R:15 s; BW
	Superman	2 × 5 + 5; R15 s; BW	Gluteus mobility	2 × 5 + 5; R15 s; BW
	Trunk mobility	2 × 5 + 5; R15 s; BW	Dead bug	2 × 5 + 5; R15 s; BW
	Bilateral jumps	2 × 10; R:20 s; BW	One-leg balance	2 × 10; R:20 s; BW
	Squats	2 × 10; R:30 s; BW	Single-leg squat 90°	2 × 10; R:30 s; BW
	Biceps curl	2 × 12 + 12; R:30 s; 3 Kg	Burpees	2 × 10; R:30 s, BW
	Lateral planks	2 × 12 + 12; R:30 s; BW	Circuit	
			20 m forward run, 20 m backpedal run, and 30 s front planks	
			2 sets, R:30 s	
Week 2	Hip mobility	2 × 12; R:15 s; BW	Hamstring mobility	2 × 8 + 8; R:15 s; BW
	Superman	2 × 8 + 8; R15 s; BW	Gluteus mobility	2 × 8 + 8; R15 s; BW
	Trunk mobility	2 × 8 + 8; R15 s; BW	Dead bug	2 × 8 + 8; R15 s; BW
	Bilateral jumps	2 × 12; R:20 s; BW	One-leg balance	2 × 12; R:20 s; BW
	Squats	2 × 15; R:30 s; BW	Single-leg squat 90°	2 × 12; R:30 s; BW
	Biceps curl	2 × 15 + 15; R:30 s; 3 Kg	Burpees	2 × 12; R:30 s, BW
	Lateral planks	2 × 15 + 15; R:30 s; BW	Circuit	
			20 m forward run, 20 m backpedal run, and 30 s front planks	
			3 sets R:30 s	
Week 3	Hip mobility	3 × 12; R:15 s; BW	Hamstring mobility	3 × 8 + 8; R:15 s; BW
	Superman	3 × 8 + 8; R15 s; BW	Gluteus mobility	3 × 8 + 8; R15 s; BW
	Trunk mobility	3 × 8 + 8; R15 s; BW	Dead bug	3 × 8 + 8; R15 s; BW
	Bilateral jumps	3 × 12; R:20 s; BW	One-leg balance	3 × 12; R:20 s; BW
	Squats	3 × 15; R:30 s; BW	Single-leg squat 90°	3 × 12; R:30 s; BW
	Biceps curl	3 × 15 + 15; R:30 s; 3 Kg	Burpees	3 × 12; R:30 s; BW
	Lateral planks	3 × 15 + 15; R:30 s; BW	Circuit	
			30 m forward run, 30 m backpedal run and 30 s front planks	
			2 sets, R:30 s	
Week 4	Hip mobility	3 × 12; R:15 s; BW	Hamstring mobility	3 × 8 + 8; R:15 s; BW
	Superman	3 × 8 + 8; R15 s; BW	Gluteus mobility	3 × 8 + 8; R15 s; BW
	Trunk mobility	3 × 8 + 8; R15 s; BW	Dead bug	3 × 8 + 8; R15 s; BW
	Bilateral jumps	3 × 12; R:20 s; BW	One-leg balance	3 × 12; R:20 s; BW
	Squats	3 × 15; R:30 s; BW	Single-leg squat 90°	3 × 12; R:30 s; BW
	Biceps curl	3 × 15 + 15; R:30 s; 5 Kg	Burpees	3 × 12; R:30 s; BW
	Lateral planks	3 × 15 + 15; R:30 s; BW	Circuit	
			30 m forward run, 30 m backpedal run, and 30 s front planks	
			3 sets, R:30 s	

R: recovery; BW: bodyweight; R: recovery; s: seconds; Kg: kilograms.

**Table 2 healthcare-13-01372-t002:** Differences in physical fitness outcomes by groups.

Groups	Metric	Baseline Mean (SD)	CI 95%	Final Assessment Mean (SD)	CI 95%	t	*p*	d	Interpretation
G1exp	Lower-body power (m)	1.36 (0.30)	1.25–1.48	1.56 (0.27)	1.46–1.67	−7.04	<0.001 **	−1.30	Large
	Upper-body power (m)	3.80 (0.86)	3.47–4.13	4.09 (0.91)	3.75–4.44	−5.94	<0.001 **	−1.10	Large
	Abdominal endurance strength (reps)	18.59 (4.18)	17.0–20.2	21.24 (4.34)	19.6–22.9	−9.72	<0.001 **	−1.80	Large
	Sprint (s)	11.30 (1.01)	10.6–11.7	10.82 (0.85)	10.5–11.1	5.22	<0.001 **	0.96	Large
G2exp	Lower-body power (m)	1.58 (0.40)	1.43–1.72	1.67 (0.39)	1.53–1.81	−5.19	<0.001 **	−0.91	Large
	Upper-body power (m)	4.73 (1.27)	4.27–5.19	4.92 (1.37)	4.43–5.42	−3.52	<0.001 **	−0.62	Moderate
	Abdominal endurance strength (reps)	19.50 (6.86)	17.0–22.0	22.84 (5.60)	20.8–24.9	−4.75	<0.001 **	−0.84	Large
	Sprint (s)	11.28 (1.22)	10.8–11.7	10.65 (1.15)	10.2–11.1	5.90	<0.001 **	1.04	Large
G1con	Lower-body power (m)	1.43 (0.27)	1.33–1.53	1.44 (0.24)	1.34–1.53	−0.76	0.448	−0.14	Small
	Upper-body power (m)	4.23 (0.83)	3.91–4.54	4.13 (0.77)	3.83–4.42	1.55	0.131	0.28	Small
	Abdominal endurance strength (reps)	15.79 (3.77)	14.4–17.2	17.17 (3.52)	15.8–18.5	−5.21	<0.001 **	−0.96	Large
	Sprint (s)	12.17 (1.14)	11.7–12.6	12.08 (1.08)	11.7–12.5	1.83	0.077	0.34	Small
G2con	Lower-body power (m)	1.64 (0.38)	1.50–1.78	1.69 (0.39)	1.54–1.83	−1.72	0.095	−0.31	Small
	Upper-body power (m)	5.42 (1.16)	4.99–5.84	5.56 (1.33)	5.08–6.05	−1.65	0.107	−0.29	Small
	Abdominal endurance strength (reps)	20.19 (7.75)	17.3–23.0	20.74 (6.99)	18.2–23.3	−0.67	0.505	−0.12	Small
	Sprint (s)	11.04 (1.28)	10.6–11.5	10.86 (1.24)	10.40–11.30	1.82	0.078	0.32	Small

CI = confidence interval; t: *t*-test paired samples; *p*: significance index (**: *p* ≤ 0.01); d: Cohen’s d; interpretation: small (d: 0.00 to ≤0.49), medium (d: 0.50 to ≤0.79), and large (d: ≥0.80).

**Table 3 healthcare-13-01372-t003:** Differences in physical fitness outcomes between groups.

Metric	Comparisons	Baseline			Final Assessment		
		Mean Differences	t	*p*	Mean Differences	t	*p*
Lower-body power	G1exp vs. G2exp	−0.21	−2.42	0.07	−0.10	−1.20	0.63
	G1exp vs. G1con	−0.06	−0.74	0.88	0.12	1.40	0.50
	G1exp vs. G2con	−0.27	−3.11	0.01 *	−0.12	−1.39	0.50
	G2exp vs. G1con	0.14	1.66	0.34	0.22	2.64	0.04 *
	G2exp vs. G2con	−0.06	−0.73	0.88	−0.01	−0.21	0.99
	G1con vs. G2con	−0.21	−2.36	0.09	−0.24	−2.82	0.02 *
Upper-body power	G1exp vs. G2exp	−0.92	−3.72	0.005 **	−0.82	−2.83	0.02 *
	G1exp vs. G1con	−0.42	−1.53	0.42	−0.03	−0.10	1.00
	G1exp vs. G2con	−1.61	−5.91	<0.001 **	−1.47	−4.98	<0.001 **
	G2exp vs. G1con	0.50	1.85	0.25	0.79	2.71	0.03 *
	G2exp vs. G2con	−0.68	−2.58	0.05	−0.64	−2.23	0.12
	G1con vs. G2con	−1.19	−4.35	<0.001 **	−1.43	−4.87	<0.001 **
Abdominal endurance strength	G1exp vs. G2exp	−0.91	−0.59	0.93	−1.60	−1.17	0.64
	G1exp vs. G1con	2.79	1.78	0.28	4.07	2.91	0.02 *
	G1exp vs. G2con	−1.60	−1.04	0.72	0.49	0.36	0.98
	G2exp vs. G1con	3.71	2.43	0.07	5.67	4.15	<0.001 **
	G2exp vs. G2con	−0.69	−0.46	0.96	2.10	1.56	0.40
	G1con vs. G2con	−4.40	−2.85	0.02 *	−3.57	−2.59	0.05 *
Sprint	G1exp vs. G2exp	0.02	0.07	1.00	0.17	0.60	0.93
	G1exp vs. G1con	−0.86	−2.81	0.02 *	−1.26	−4.36	<0.001 **
	G1exp vs. G2con	0.26	0.87	0.82	−0.03	−0.13	0.99
	G2exp vs. G1con	−0.89	−2.95	0.02 *	−1.43	−5.07	<0.001 **
	G2exp vs. G2con	0.24	0.81	0.84	−0.20	−0.75	0.87
	G1con vs. G2con	1.13	−3.73	0.002 **	1.22	4.30	<0.001 **

t: Tukey post hoc; *p*: significance index (*: *p* ≤ 0.05; **: *p* ≤ 0.01).

## Data Availability

The data that support the findings of this study are available on request from the corresponding author. The data are not publicly available due to privacy or ethical restrictions.

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
