# Peer review of "An Integrative Neuromuscular Training Program in Physical Education Classes Improves Strength and Speed Performance"

_healthcare, 2025, doi:10.3390/healthcare13121372_

Round 1
Reviewer 1 Report
Comments and Suggestions for Authors
Congratulations on your article,
It is very well developed and with a very interesting subject matter.
Within it I have attached minor revisions that in my view will improve the clarity of the article.
Abstract
Line 19- Consider replacing "academic ages" with "school grades" or "age groups" for clarity.
Line 21- "A total of 121 students, aged 11–12 (G1) and 15–16 (G2)...”
Line 26- "Experimental groups showed significant improvements (p<0.001) in lower body power...".Although correct, it would be useful to introduce ‘compared to controls’ here to emphasise the comparative design.
Line 30- "A comparison by age group revealed significantly greater improvements in upper body power in G1 vs G2 (t:-2.83; p=0.2)". The value of p=0.2 is not statistically significant. This statement should be revised or qualified.
Introduction
Line 37-“...is a crucial research area...”
Line 43- “…fails to meet current…”
Methods
Line 85- "...no previous participation in an INT program and perform all the intervention classes...". “…and ability to perform all the intervention classes…”
Line 89-"...energy drinks, caffeine, etc.". In scientific texts it is better to avoid ‘etc.’; specify if necessary or delete it.
Line 91- In the participants, why don't you describe how many students are in each level? This would strengthen the sample
Results
Line 174- "All variables followed normal distribution and were homogeneous." Consider modifying it to: “All variables were normally distributed and showed homogeneity of variance.”
Line 182 (Table 2)- The table is clear, but it is recommended to use letters (a, b, c...) to indicate significant differences between groups.
Discussion
Line 214- "...regardless of their academic grade level." Better “...regardless of whether they were in preadolescence or adolescence.”
General
There are several repetitions of the term ‘physical education classes (PEC)’ and ‘integrative neuromuscular training (INT)’. Once defined, only the acronym can be used to avoid redundancy.
In academic English, it is recommended to avoid expressions such as ‘on the other hand’ without first using ‘on the one hand’.
Check consistency in the use of commas, hyphens and abbreviations (e.g. ‘d-Cohen’ vs. ‘Cohen's d’).
This study evaluated the effects of an integrative neuromuscular training (INT) programme implemented during physical education (PE) classes in Spanish secondary school students of two age groups (11-12 and 15-16 years old). A total of 121 students were randomly assigned to four groups (two experimental and two controls), and the INT programme consisted of two sessions per week for four weeks. Lower and upper body power, abdominal strength and speed were measured. The experimental groups showed significant improvements over the control groups, especially in abdominal strength and speed.
Limited justification for the intervention period
Although significant improvements are obtained in four weeks, the study does not provide a clear rationale as to why such a short intervention was chosen, nor does it discuss possible effects of a longer duration. It is recommended that the choice of the period and its potential impact be discussed.
Absence of baseline physical activity assessment (baseline PA)
No objective or subjective measurement of participants' previous level of physical activity was included. This is relevant as it may condition the magnitude of the observed improvements. This factor should be considered as a limitation or incorporated in future studies.
Lack of control of the teacher effect
The manuscript does not clarify whether the sessions were taught by the same person or whether there was variability between groups. This could introduce bias. It is recommended to clearly describe who delivered the sessions and whether there was any standardisation.
Confusing statistical analysis in some parts
The tables mix intra- and inter-group effects, but in some post hoc contrasts it is not clear whether correction for multiple comparisons (e.g. Bonferroni, Tukey) was applied. In addition, there are relevant p-values that appear as 0.07 or similar without discussion of their practical relevance.
In conclusion, the study addresses a relevant issue from a practical and innovative perspective (use of INT in EF classes), with an adequate sample and promising findings. However, it requires methodological improvements and clarity in writing, as well as further discussion of limitations, duration of intervention and generalisability of results. A careful revision of style and statistics, together with methodological clarifications, would significantly improve the quality of the manuscript.
Reviewer 2 Report
Comments and Suggestions for Authors
Review Report
The manuscript presents a valuable and relevant study investigating the impact of an Integrative Neuromuscular Training (INT) program on strength and sprint performance among preadolescent and adolescent male students within a school-based physical education setting.
However, several important improvements are required in the abstract, introduction, and methods sections to strengthen the clarity, scientific rigor, and presentation quality of the manuscript.
Detailed suggestions for improvement
- Abstract -the abstract follows a logical structure, but it lacks specific methodological and contextual information. The design of the study (e.g., randomized controlled trial or quasi-experimental design) is not explicitly mentioned. The total sample size should be reported upfront for clarity. One of the reported p-values (“p = 0.2”) appears to be a typographical error, as it would not typically indicate significance in this context. Please review and correct if necessary. The abstract would benefit from a brief statement on the practical significance of the findings (e.g., application in school curricula or long-term health implications).
- Introduction -the introduction provides general background on physical activity and youth health but lacks a clearly defined research gap. It would benefit from a stronger justification for the study, such as: limited existing research on school-based INT programs targeting both preadolescents and adolescents., lack of studies comparing responses across different age groups during adolescence. The aim of the study should be clearly stated in the final paragraph of the introduction.
- Methods -the methodology is generally well-structured, with clear descriptions of inclusion criteria, training program, and performance assessments. However, several clarifications are needed: randomization procedure (Was it random assignment? If so, how was it done (e.g., computer-generated, envelope method)? Test reliability - please include evidence or citations for the reliability and validity of the physical performance tests used (e.g., sprint test, abdominal strength test). Blinding - Were test administrators blinded to group allocation to reduce bias?
4.In the statistical analysis section:
-specify the alpha level used (e.g., p < 0.05);
-indicate whether any corrections for multiple comparisons were applied;
-ensure clarity in terms like “BW,” “R,” and “Kg” by defining them in Table 1;
-improve the clarity of Figures 1 and 2 by increasing resolution and adding axis units, including asterisks or symbols to highlight statistically significant changes and expanding captions to identify the age groups for G1exp and G2exp.
5.Discussion section -the discussion section interprets the findings appropriately, with references to relevant literature. However, the practical implications could be more explicitly addressed (e.g., feasibility of implementing INT in schools, teacher training needs). Please expand on the developmental differences between age groups in response to INT, focusing on maturational factors and neuromuscular plasticity.
6.Limitations -the authors should clearly state whether the study included only male participants, and note this as a limitation regarding generalizability. The absence of follow-up data to assess long-term impact should also be acknowledged.
Ensure consistency in tense and terminology throughout the manuscript!
Reviewer 3 Report
Comments and Suggestions for Authors
An Integrative Neuromuscular Training Program in Physical 2 Education Classes Improves Strength and Speed Performance
Review report
Dear authors,
Firstly, I would like to extend my congratulations on your endeavors to enhance the quality of physical education instruction in schools. It is crucial to acknowledge the significance of initiatives that promote the well-being of young individuals. These endeavors should be endorsed by both the pertinent ministries and the state in its entirety. This critical approach to research is undertaken with the objective of presenting research findings to the scientific community in the most optimal manner.
Abstract
Lines 24 to 25. Comment 1: Agility, in the context of physical fitness, can be defined as the ability to quickly and effectively change body position while maintaining control. This involves a combination of speed, balance, coordination, reflexes, and strength. It's about reacting to changing situations and adapting movement accordingly. Upon reading the abstract, the reader may naturally inquire into the rationale behind training agility and speed as discrete components within the context of a physical education class, given the potential for these skills to be intertwined. I would appreciate your feedback on this matter.
Introduction
Lines 39 to 41. Comment 2: It is acknowledged that the parameters delineated in the designated rows accurately measure physical activity. However, the term "physical activity" also encompasses other activities of daily life, such as housework, gardening, and any manual activity. The term "physical activity" is often perceived as synonymous with "physical exercise," which refers to a structured series of movements that is initiated and concluded at specific points. It would be beneficial to provide a more precise definition of the term "physical activity."
Line 54. Comment 3: Could you please provide a concise definition of the term "healthier lifestyle" that extends beyond the literature you have cited?
Lines 78 to 79. Comment 4: It is recommended that more specific research hypotheses be formulated, as this would serve to strengthen the study. It is imperative that more precise hypotheses be formulated.
Materials and Methods
Line 80. Comment 5: It is my estimation that the addition of a heading such as "Study Design" in the initial paragraph would enhance the practicality of the study.
Lines 88 to 90. Comment 6: An elucidation of the exclusion criteria for participants would be greatly appreciated. It is imperative to examine the question of whether an 11-year-old child can consume caffeine or energy drinks. It is imperative to ascertain whether these criteria were intended for older children.
Lines 115 to 118. Comment 7: Please correct the following “After the warm-up, students performed the following assessments in this order. with a two-minute rest between each test:”. Also “These test are widely 117 used in the literature due to their strong validity and reliability 29”
Lines 139 to 141. Comment 8: You wrote “Sprint: 4x10 meters.
The student stood in front of the starting line and completed a series of round-trip sprints over 10-meter distance, totaling 80 meters at the highest possible speed.”. Would you please add to your description how does a 4X10 meter speed test result in 80 meters of distance being completed?
Line 153, Table 1. Comment 9: Please clarify what is meant by the term “Single leg squat”? Does the exercise “pistol squat”?
Discussion
Lines 211 to 212. Comment 10: Do students aged 11-12 go to primary or secondary school? Please clarify.
Discussion general comment: In order to facilitate comprehension of the present discussion, it would be advisable to present the results in a sequential manner, followed by a thorough examination of the disparities that emerged as a consequence of the intervention between each group (G1exp and G2exp). Subsequent to this, a comparative analysis between the groups should be conducted, given the rationality of contrasting children (11-12 years old) with adolescents (15-16 years old). Subsequently, for each comparison, the extant literature that aligns or contrasts with our results is cited, and the potential for similarities or differences is elucidated. Upon reading the discussion, it was difficult to ascertain the specific elements that were of importance.
Reviewer 4 Report
Comments and Suggestions for Authors
Thank you for the opportunity to review your manuscript. This study makes a valuable contribution to the field of physical education and youth fitness. Below are several suggestions to improve clarity, methodological transparency, and the contextual framework:
General Comments:
- The manuscript is well-structured and the research question is clearly defined. The intervention is well-executed and the findings are relevant.
- Consider discussing the generalizability of your findings beyond the studied sample, especially in relation to different cultural, socio-economic, or school contexts.
Specific Comments:
Title:
- Consider specifying the population (e.g., Spanish adolescents) in the title to improve clarity and discoverability.
Abstract:
- Include effect sizes or confidence intervals alongside p-values for key outcomes.
- Clarify in the conclusion sentence that improvements were seen in both age groups, but that preadolescents had greater sensitivity to training.
Introduction:
- Lines 50–54: Expand on how the pediatric triad affects functional outcomes in youth.
- Clearly justify why the chosen INT protocol (frequency, duration, and content) is expected to be effective based on prior studies.
Methods:
- Lines 85–89: The inclusion/exclusion criteria could be better structured and more precisely worded. Clarify whether previous exposure to structured training was controlled.
- Table 1 is detailed and useful, but a brief synthesis in the text of the progression in exercise load and complexity would help the reader interpret its structure.
- Clarify if the same instructor conducted both the PE and INT sessions, and whether blinding was attempted during assessments.
Results:
- Figures 1 and 2 are informative, but it would be helpful to add a brief explanatory legend to each.
- In Table 3, include confidence intervals for mean differences.
- Clarify the use of “G3” and “G4” in the text (line 197), which do not appear to be defined elsewhere.
Discussion:
- The discussion is comprehensive and well-referenced. Consider adding a section on practical implementation barriers (e.g., school resources, staff training).
- Line 321–325: Expand on the developmental differences between groups and potential reasons why adolescents showed less change than preadolescents.
- Highlight the clinical or public health relevance of the findings more clearly at the end of the discussion.
Conclusion:
- The conclusion is concise and well-aligned with the findings. You might suggest specific recommendations for future studies (e.g., longer follow-ups, sex-specific responses, or functional health markers).
Tables and Figures:
- Ensure all tables use consistent decimal formatting and align CI notation.
- Add legends to figures explaining statistical significance indicators (*, **).
Any papers recommended in the report are for reference only. They are not mandatory. You may cite and reference other papers related to this topic.
Reference Enhancements:
- Include "Osteoarthritis: a call for research on central pain mechanism and personalized prevention strategies" Relevant for supporting the need for preventive approaches and individualized training protocols, which may also be applicable to younger populations.
- “Effects of Orthopedic Manual Therapy on Pain Sensitization in Patients with Chronic Musculoskeletal Pain: An Umbrella Review with Meta-Meta-Analysis” Strengthens the scientific rationale for the role of exercise and manual therapy in modulating central pain mechanisms and promoting functional adaptation.
- The manuscript demonstrates good command of academic English. Minor revisions in grammar and syntax are needed, particularly in prepositions and verb tenses (e.g., line 40 “Physical activity… is crucial research area” → a crucial area of research).
- Some inconsistencies in spacing and punctuation should be corrected during proofreading (e.g., commas before citations, spacing between references and parentheses).
Round 2
Reviewer 1 Report
Comments and Suggestions for Authors
Congratulations on your article.
You have responded correctly to all the comments attached to it.
Author Response
Thank you very much por your contributions!
Reviewer 2 Report
Comments and Suggestions for Authors
Thank you for your efforts in addressing the initial round of reviewer comments. The revised version of the manuscript has shown substantial improvements in terms of study design clarity, methodological transparency, and presentation of key findings.
Several issues remain!
While the physical fitness assessments are well described, there is still no citation or supporting reference regarding the validity and reliability of the tests employed (e.g., standing long jump, 30-second crunches, medicine ball throw, sprint test). Given that these measures are central to the outcomes assessed, including references from peer-reviewed literature confirming their psychometric properties is essential to support the robustness of your findings.
Some improvement was noted in the graphical presentation of Figures 1 and 2, these figures still fall short of publication quality. I recommend that you enhance the visual clarity of the figures by using high-resolution graphics, clearly labeling all axes, and visually marking statistically significant changes with asterisks or symbols. Expanded figure captions should also specify the age groups and clarify the nature of the changes illustrated.
The discussion section would benefit from a more detailed exploration of age-related differences in response to INT, particularly with respect to maturational status, neuromuscular plasticity, and developmental physiology. While your analysis highlights differential outcomes between groups, further interpretation of why these differences may have occurred—grounded in current literature—would add valuable depth and context to your discussion.
The limitations section needs to be explicitly strengthened. If the sample consisted exclusively of male participants, this must be clearly stated along with a reflection on how this affects the generalizability of your findings. The absence of follow-up data to determine the long-term sustainability of the intervention effects should be acknowledged as a limitation of the current design.
Author Response
Thank you for your efforts in addressing the initial round of reviewer comments. The revised version of the manuscript has shown substantial improvements in terms of study design clarity, methodological transparency, and presentation of key findings.
Several issues remain!
While the physical fitness assessments are well described, there is still no citation or supporting reference regarding the validity and reliability of the tests employed (e.g., standing long jump, 30-second crunches, medicine ball throw, sprint test). Given that these measures are central to the outcomes assessed, including references from peer-reviewed literature confirming their psychometric properties is essential to support the robustness of your findings.
Line 135-13. These tests are widely used in the literature due to their strong validity and reliability
Reference:
Tabacchi, G.; López-Sánchez, G.; Åžahin, F. N.; Kızılyallı, M.; Genchi, R.; Basile, M.; Kirkar, M.; Silva, C.; Loureiro, N.; Teixeira, E.; Demetriou, Y.; Sturm, D.; Pajaujiene, S.; Zuoziene, I.; Gómez-López, M.; RaÄ‘a, A.; Paušić, J.; Lakicevic, N.; Petrigna, L.; Bianco, A. Field-Based Tests for the Assessment of Physical Fitness in Children and Adolescents Practicing Sport: A Systematic Review within the ESA Program. Sustainability 2019, 11, 7187. https://doi.org/10.3390/su11247187
Some improvement was noted in the graphical presentation of Figures 1 and 2, these figures still fall short of publication quality. I recommend that you enhance the visual clarity of the figures by using high-resolution graphics, clearly labeling all axes, and visually marking statistically significant changes with asterisks or symbols. Expanded figure captions should also specify the age groups and clarify the nature of the changes illustrated.
We have improved the resolution of the images. The significant differences are marked with asterisks on the bars, and we have included the age groups in the figure legends.
The discussion section would benefit from a more detailed exploration of age-related differences in response to INT, particularly with respect to maturational status, neuromuscular plasticity, and developmental physiology. While your analysis highlights differential outcomes between groups, further interpretation of why these differences may have occurred—grounded in current literature—would add valuable depth and context to your discussion.
We added this paragraph: Upon observing the results in the experimental groups (G1exp and G2exp), significant improvements are seen in all areas except abdominal endurance strength (G1exp), with large effect sizes. This implies that the INT programs provide substantial quantitative and qualitative benefits in physical fitness and health for young individuals, regardless of their age group.
We added: “The G1exp group shows larger effect sizes than the G2exp group. Puberty and the prepubertal stage exhibit greater neuroendocrine and likely, neuromuscular plasticity, requiring less stimulus to induce changes compared to late adolescence. Interventions during these stages may be more effective due to this increased sensitivity.”
Myer, G. D.; Faigenbaum, A. D.; Edwards, N. M.; Clark, J. F.; Best, T. M.; Sallis, R. E. Sixty Minutes of What? A Developing Brain Perspective for Activating Children with an Integrative Exercise Approach. British Journal of Sports Medicine 2015. https://doi.org/10.1136/bjsports-2014-093661
Myer, G. D.; Faigenbaum, A. D.; Ford, K. R.; Best, T. M.; Bergeron, M. F.; Hewett, T. E. When to Initiate Integrative Neuromuscular Training to Reduce Sports-Related Injuries and Enhance Health in Youth? Current Sports Medicine Reports 2011, 10 (3), 155. https://doi.org/10.1249/JSR.0b013e31821b1442
The limitations section needs to be explicitly strengthened. If the sample consisted exclusively of male participants, this must be clearly stated along with a reflection on how this affects the generalizability of your findings. The absence of follow-up data to determine the long-term sustainability of the intervention effects should be acknowledged as a limitation of the current design.
We added this: Additionally, it would have been interesting to follow up on the results to determine the long-term effect of the findings and/or implement a counterbalancing.
We added this: Despite knowing the benefits of INT in these population groups, some practical challenges should be mentioned that professionals must consider: 1) structuring their sessions to include INT programs with a time interval of 10-15 minutes, 2) understanding the material limitations that may affect exercise progression, 3) in educational institutions, it would be necessary to provide training for safe and effective implementation, and 4) considering the maturation and developmental status of the participants.
Reviewer 3 Report
Comments and Suggestions for Authors
Review round 2
Article
An Integrative Neuromuscular Training Program in Physical Education Classes Improves Strength and Speed Performance
Dear authors, I would like to express my gratitude for your meticulous revisions, which have addressed the previous feedback with such care and consideration. The present study offers significant insights into the efficacy of an Integrative Neuromuscular Training (INT) program in enhancing strength and sprint performance among youth. The modifications have led to enhanced methodological clarity and scientific transparency. The following recommendations are provided to further refine the manuscript:
Line 112. Comment 1: The study does not specify how the sample size was determined or whether a power analysis was conducted. The provision of a concise rationale, substantiated by a G-power analysis, would serve to enhance methodological transparency.
Line 206. Comment 2: Clarification of Between-Group Comparisons. The manuscript presents several statistical comparisons; however, it is unclear whether interactions between age groups were tested. In the absence of such analysis, it would be advantageous to deliberate on the potential disparities in training adaptations between preadolescents and adolescents.
Line 236 to the end of discussion. Comment 3: Effect Size Interpretation. While Cohen's d effect sizes are reported, there is an absence of any discussion regarding their practical significance. A brief exposition of these effect sizes in the context of real-world enhancements would further enrich the findings.
Lines 270 to 277. Comment 4: Moderating Claims on Training Effects. The manuscript suggests that the INT intervention effectively improves strength and sprint performance across all age groups. However, the results of the study indicated that the older control group (G2con) did not demonstrate significant improvements, suggesting the presence of potential age-related variability in training adaptations. In order to facilitate a balanced discussion, it is recommended that the authors acknowledge that younger participants may benefit more due to higher neuromuscular plasticity, while older adolescents may require longer or more intense training stimuli to achieve similar effects. A brief clarification in the discussion would improve the study's accuracy and assist in ensuring that its conclusions are precise.
Lines 359 to 363. Comment 5: Practical Implementation Considerations. The integration of Integrative Neuromuscular Training (INT) into physical education sessions is a recommended practice that has been thoroughly substantiated by research. However, the manuscript does not address potential concerns regarding feasibility, including time constraints, the necessity for teacher training, and equipment availability in school settings. A brief mention of these practical challenges, along with possible solutions or adaptations, would strengthen the applicability of the findings and ensure realistic implementation in educational environments.
In sum, the revised manuscript exhibits notable scientific contributions. Addressing these final refinements will further enhance the clarity, methodological rigor, and practical relevance of the study.

Author Response
Dear authors, I would like to express my gratitude for your meticulous revisions, which have addressed the previous feedback with such care and consideration. The present study offers significant insights into the efficacy of an Integrative Neuromuscular Training (INT) program in enhancing strength and sprint performance among youth. The modifications have led to enhanced methodological clarity and scientific transparency. The following recommendations are provided to further refine the manuscript:
Line 112. Comment 1: The study does not specify how the sample size was determined or whether a power analysis was conducted. The provision of a concise rationale, substantiated by a G-power analysis, would serve to enhance methodological transparency.
We added: “The statistical power of the study was assessed using G*Power (version 3.1.9.7) for an analysis of variance (ANOVA) with a moderate effect size (Cohen's d = 0.50), a significance level (α) of 0.05, and a desired power of 0.80. With a total of 121 participants distributed across four groups, the calculated statistical power was 99.98%, ensuring a high probability of identifying a moderate effect if present.”
Line 206. Comment 2: Clarification of Between-Group Comparisons. The manuscript presents several statistical comparisons; however, it is unclear whether interactions between age groups were tested. In the absence of such analysis, it would be advantageous to deliberate on the potential disparities in training adaptations between preadolescents and adolescents.
Thank you for your feedback. We believe that in the results section, we should not deliberate on the findings, and it might be more appropriate to do so in the discussion section.
Line 236 to the end of discussion. Comment 3: Effect Size Interpretation. While Cohen's d effect sizes are reported, there is an absence of any discussion regarding their practical significance. A brief exposition of these effect sizes in the context of real-world enhancements would further enrich the findings.
We added this paragraph: Upon observing the results in the experimental groups (G1exp and G2exp), significant improvements are seen in all areas except abdominal endurance strength (G1exp), with large effect sizes. This implies that the INT programs provide substantial quantitative and qualitative benefits in physical fitness and health for young individuals, regardless of their age group.
Lines 270 to 277. Comment 4: Moderating Claims on Training Effects. The manuscript suggests that the INT intervention effectively improves strength and sprint performance across all age groups. However, the results of the study indicated that the older control group (G2con) did not demonstrate significant improvements, suggesting the presence of potential age-related variability in training adaptations. In order to facilitate a balanced discussion, it is recommended that the authors acknowledge that younger participants may benefit more due to higher neuromuscular plasticity, while older adolescents may require longer or more intense training stimuli to achieve similar effects. A brief clarification in the discussion would improve the study's accuracy and assist in ensuring that its conclusions are precise.
We added: “The G1exp group shows larger effect sizes than the G2exp group. Puberty and the prepubertal stage exhibit greater neuroendocrine and likely, neuromuscular plasticity, requiring less stimulus to induce changes compared to late adolescence. Interventions during these stages may be more effective due to this increased sensitivity.”
Myer, G. D.; Faigenbaum, A. D.; Edwards, N. M.; Clark, J. F.; Best, T. M.; Sallis, R. E. Sixty Minutes of What? A Developing Brain Perspective for Activating Children with an Integrative Exercise Approach. British Journal of Sports Medicine 2015. https://doi.org/10.1136/bjsports-2014-093661
Myer, G. D.; Faigenbaum, A. D.; Ford, K. R.; Best, T. M.; Bergeron, M. F.; Hewett, T. E. When to Initiate Integrative Neuromuscular Training to Reduce Sports-Related Injuries and Enhance Health in Youth? Current Sports Medicine Reports 2011, 10 (3), 155. https://doi.org/10.1249/JSR.0b013e31821b1442
Lines 359 to 363. Comment 5: Practical Implementation Considerations. The integration of Integrative Neuromuscular Training (INT) into physical education sessions is a recommended practice that has been thoroughly substantiated by research. However, the manuscript does not address potential concerns regarding feasibility, including time constraints, the necessity for teacher training, and equipment availability in school settings. A brief mention of these practical challenges, along with possible solutions or adaptations, would strengthen the applicability of the findings and ensure realistic implementation in educational environments.
We added this: Despite knowing the benefits of INT in these population groups, some practical challenges should be mentioned that professionals must consider: 1) structuring their sessions to include INT programs with a time interval of 10-15 minutes, 2) understanding the material limitations that may affect exercise progression, 3) in educational institutions, it would be necessary to provide training for safe and effective implementation, and 4) considering the maturation and developmental status of the participants.
In sum, the revised manuscript exhibits notable scientific contributions. Addressing these final refinements will further enhance the clarity, methodological rigor, and practical relevance of the study.
Reviewer 4 Report
Comments and Suggestions for Authors
The revised manuscript is clear, methodologically sound, and presents its findings in a well-structured and coherent manner. The authors have adequately addressed the objectives of the study, and no additional changes are requested. I recommend the article for publication in its current form.
Comments on the Quality of English LanguageThe manuscript is written in fluent and professional academic English. The text reads smoothly, and the terminology used is appropriate for the field. No language revisions are necessary at this time.
Author Response

(The authors gave the same response as above.)
